# Comprehensive *in silico* analyses of fifty-one uncharacterized proteins from *Vibrio cholerae*

Sritapa Basu Mallick[1], Sagarika Das[1¤], Aravind Venkatasubramanian[1], Sourabh Kundu[2], Partha Pratim Datta[1] *

1 Department of Biological Sciences, Indian Institute of Science Education and Research Kolkata, Mohanpur, WB, India, 2 Ramakrishna Mission and Vivekananda Educational and Research Institute, Narendrapur, Kolkata, WB, India

¤ Current address: Department of Laboratory Medicine, Institute of Biomedicine, University of Gothenburg, Gothenburg, Sweden
* partha_datta@iiserkol.ac.in

**Data Availability Statement:** All relevant data are within the manuscript and its Supporting Information files.

## Abstract

Due to the rise of multidrug-resistant strains of *Vibrio cholerae* and the recent cholera outbreaks in African and Asian nations, it is imperative to identify novel therapeutic targets and possible vaccine candidates. In this regard, this work primarily aims to identify and characterize new antigenic molecules using comparative RNA sequencing data and label-free proteomics data, carried out with essential GTPase *cgtA* knockdown and wild-type strain of *V. cholerae*. We identified hitherto 51 characterized proteins from high-throughput RNA-sequencing and proteomics data. This work involved the assessment of their physicochemical characteristics, subcellular localization, solubility, structures, and functional annotations. In addition, the immunoinformatic and reverse vaccinology technique was used to find new vaccine targets with high antigenicity, low allergenicity, and low toxicity profiles. Among the 51 proteins, 24 were selected based on their immunogenic profiles to identify B/T-cell epitopes. In addition, 20 prospective therapeutic targets were identified using virulence predictions and related investigations. Furthermore, two proteins, UniProt ID- Q9KRD2 and Q9KU58, with molecular weight of 92kDa and 12kDa, respectively, were chosen for cloning and expression towards *in vitro* biochemical characterization based on their range of expression patterns, high antigenic, low allergenic, and low toxicity properties. In conclusion, we believe that this study will reveal new facets and avenues for drug discovery and put us a step forward toward novel therapeutic interventions against the deadly disease of cholera.

## Introduction

*V. cholerae* is a pathogenic Gram-negative bacterium that is known to be the etiological driver of the deadly diarrheal disease cholera. Approximately 1.3–4 million people are estimated to be affected annually, and around 95,000 people die annually in 51 endemic countries due to this fatal disease [1]. Instances of cholera have been observed globally and etiologically tied to the consumption of contaminated water or seafood, which is often infested with a cocktail of

**Funding:** MoE STARS (Award number- STARS/APR2019/BS/581/FS) was used. We also thank IISER, Kolkata for partially funding and supporting PPD lab. The funders had no role in study design, data collection and analysis, decision to publish, or preparation of the manuscript.

**Competing interests:** The authors have declared that no competing interests exist

various strains of *Vibrio spp.*, such as *V. cholerae*, *V. vulnificus*, and *V. parahaemolyticus* [2]. This, coupled with the emergence of antibiotic-resistant variants of cholera [3], has driven cholera research worldwide, including in developed countries. Since we dwell in an era of multidrug resistance (MDR), there is the utmost need to explore and study new potential drug targets.

CgtA is a well-studied potential drug target [4] and is an essential ribosome-associated GTPase that is known to play a monumental role in various cellular functions, such as cell growth and DNA replication [5], chromosome partitioning [6], DNA repair [7], ribosome biogenesis [8–10] and stringent response [11]. However, the mechanism by which CgtA exerts its pleiotropic effect is still unknown. In one of our recent studies [12], high-throughput techniques like RNA-seq and labeled-free proteomics were used to conduct a comparative study between *cgtA* knockdown and wild-type *Vibrio cholerae* strain, in which we identified several uncharacterized proteins whose expression patterns were seen to be significantly altered when *cgtA* was knocked down from the *V. cholerae* genome. Understanding these biochemically, structurally and functionally uncharacterized proteins can pave the way towards mechanistic insights of how this essential GTPase, CgtA, exerts its pleotropic effect. Hence, we carried out a comprehensive *in silico* analysis of these uncharacterized proteins, which allowed us to predict their physicochemical and immunogenic properties and hypothesize and design various *in vitro* and *in vivo* experiments to characterize these proteins, which will lead us to understand the basis of cholera pathogenesis at a deeper level. Also, we have successfully identified a number of potential drug-targets and vaccine candidates among those 51 uncharacterized proteins that will facilitate the production of various vaccine constructs and drugs through *in vivo* and *in vitro* experiments against the *Vibrio cholerae* pathogen. Recently, a similar study has also been carried out using a reverse vaccinology study in *V. parahaemolyticus* [13].

In addition, we have also cloned and expressed two uncharacterized proteins, which validated our *in silico* and bioinformatics studies. **Fig 1** depicts the summary and strategic pipeline of our objectives, mission, and approaches adopted to fulfil our mission. Our study will undoubtedly reveal new facets of cholera pathogenesis and will lead us to a step forward toward novel therapeutic interventions against this disease.

## Materials and methods

### Sequence retrieval, and data collection

*V. cholerae* serotype O1 (strain ATCC 39315/El Tor Inaba N16961) was used in the present study. The complete genome sequence of *V. cholerae* serotype O1 was first reported in 2000 [14]. RNA-seq data that was used for this study were collected from the European Nucleotide Archive (ENA) at EMBL-EBI with BioProject accession number PRJEB53015 (SRA accession number ERP137772) (https://www.ebi.ac.uk/ena/browser/view/PRJE). Furthermore, the mass spectrometry proteomics data were collected from the ProteomeXchange Consortium with the data set identifier PXD034015.

### Quality control and transcriptomics data analysis

The quality of the RNA-seq data generated by Das et al., 2023 [12] was checked using FastQC and MultiQC software. The data were checked for base call quality distribution, percentage of bases above Q20 and Q30, %GC, and sequencing adapter contamination. All the samples passed the quantity control (QC) threshold (Q20 > 95%). The raw sequence reads were processed to remove adapter sequences and low-quality bases using fastp. The QC-passed reads were mapped onto the indexed *V. cholerae* reference genome (O1 biovar El Tor strain N16961) using the Bowtie 2 aligner. On average, 98.55% of the reads were aligned to the reference genome. Gene-level expression values were obtained as read counts using featureCounts

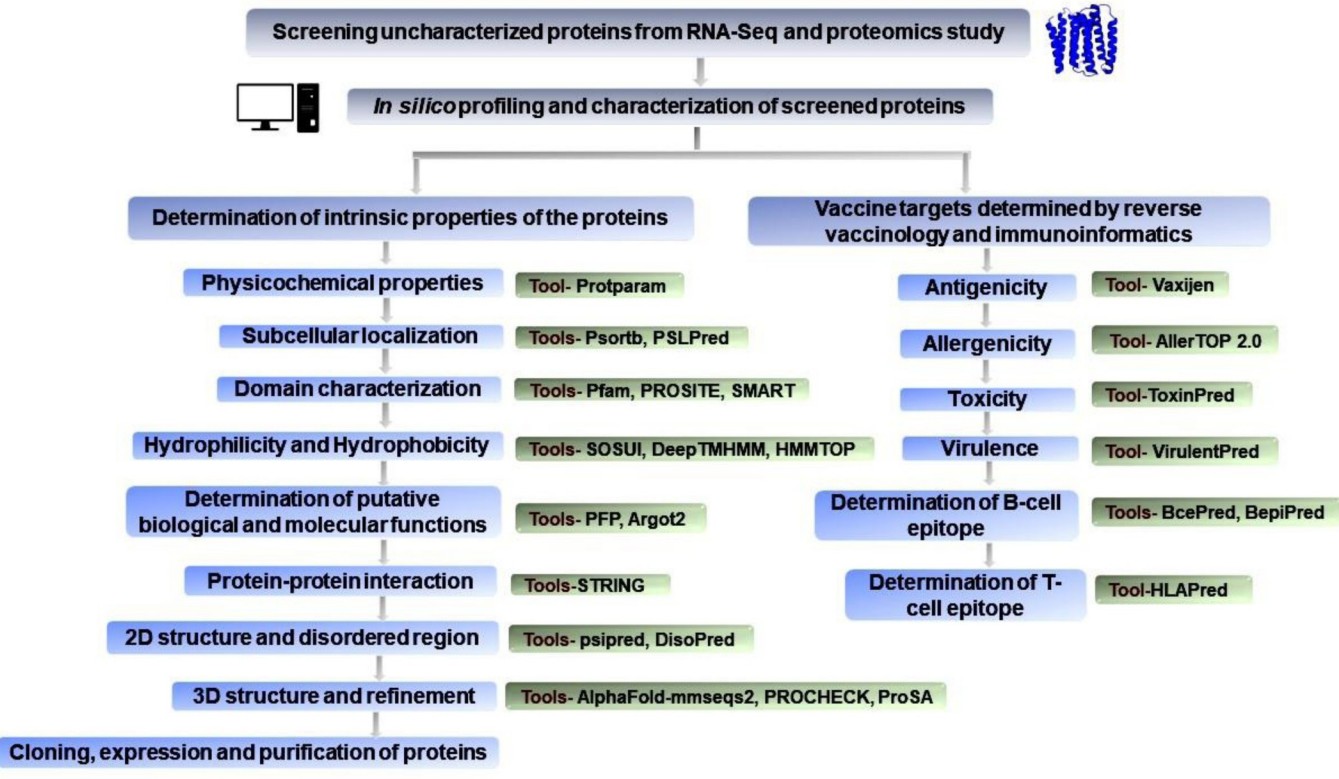

**Fig 1. Schematic workflow of the comprehensive *in silico* study of 51 uncharacterized and hypothetical proteins: The blue-colored boxes indicate the methodology and approaches that has been followed in the study.** The green-colored boxes indicate the tools that have been executed for its corresponding approach.

software. Differential expression analysis was carried out using the DESeq2 package. Thread counts were normalized (variance-stabilized normalized counts) using DESeq2, and differential enrichment analysis was performed. The test sample was compared to the control sample. Genes with an absolute $\log_2$ fold change of $\geq 1$ and a *P* value of $\leq 0.05$ were considered significant. The p-value and the fold change are noted in **S1 Table**.

### Descriptive statistics of the differentially regulated proteins identified by label-free proteomics

A label-free proteomics study was performed according to Das et al., 2023 [12]. The proteins identified by label-free proteomics were statistically validated by ANOVA. ANOVA was used to validate whether the means of more than two groups were significantly different from each other. After the ANOVA, the significantly altered proteins (up- and downregulated) were identified by the p-value. The list of altered proteins was further sorted based on the fold change (abundance ratio). The fold change refers to the ratio of the abundance of a protein under the mutant condition to that under the wild-type condition. A less stringent cutoff was applied to capture more altered proteins. The cutoff value for sorting proteins based on the fold change in the number of significantly upregulated proteins was $\geq 1.2$ ($\log_2(1.2) = 0.26$). However, for the significantly downregulated proteins, the cutoff value for fold change was

$\leq$0.83 [$\log_2(0.83) = -0.26$]. On the logarithmic scale, the significantly altered proteins with a $\log_2$ fold change between +0.26 and −0.26 were not considered for further analysis.

## Physicochemical properties and functional characterizations

The ProtParam web server by ExPASy (https://web.expasy.org/protparam/) [15] was used to identify the physicochemical characteristics of the uncharacterized proteins such as molecular weight, theoretical isoelectric point (pI), molar extinction coefficient, instability index, aliphatic index, grand average hydropathy (GRAVY), total number of positively charged (Arg +Lys) and negatively charged (Asp+Glu) residues based on their amino acid sequences (**S2 Table** and **Fig 2A**) and amino acid composition profile (%) (**S3 Table**). The molar extinction coefficient is the measure of the amount of light that proteins absorb at a specific wavelength. A high molar extinction coefficient value indicates the presence of a high concentration of cysteine, tryptophan, and tyrosine in the candidate proteins. The instability index provides an estimation of the stability of a protein in a test tube. A protein whose instability index is greater than 40 is predicted to be instable in solution. The aliphatic index of a protein is defined as the relative volume occupied by aliphatic side chain amino acids like alanine, valine, leucine, and isoleucine. It may be considered as a positive factor for the enhancement of thermo-stability of globular protein. A high aliphatic index indicates that a protein is thermo-stable over a wide temperature range. The GRAVY score for a protein is calculated as the sum of the hydropathy values of all of the amino acids divided by the number of residues in the query sequence. A low GRAVY value indicates the possibility of a protein being a globular or hydrophilic protein rather than membranous. A comprehensive analysis of these physicochemical properties will provide us an insight of the possible biological functions of these uncharacterized proteins. In addition, the predicted traits and properties like instability index and solubility will allow us to design effective strategies to express and purify these proteins for downstream biochemical and functional characterization.

## Prediction of subcellular localization

The subcellular localization of each uncharacterized protein was predicted by PSORTb version 3.0.2 (http://www.psort.org/psortb/). PSORTb is the most precise web-based tool used for the prediction of subcellular localization in bacterial protein sequences by submitting one or more Gram-positive or Gram-negative bacterial sequences or archaeal sequences in FASTA format [16]. A score is assigned for every localization site which reflects the confidence level of the prediction. A score, higher than the cut-off value of 7.5 indicates a strong confidence in the predicted localization. Further, the results were confirmed by PSLpred (https://webs.iiitd.edu.in/raghava/pslpred/submit.html), which accurately predicts the subcellular localization of uncharacterized proteins based on a hybrid approach that integrates PSI_BLAST and three SVM based on physicochemical properties, residue composition and dipeptide [17]. The overall accuracy of the prediction is expressed in terms of a percentage (**S4 Table**).

## Identification of protein motifs, domains, and families

The uncharacterized proteins were classified into families and predicted for domains using the InterPro server (https://www.ebi.ac.uk/interpro/) [18]. InterPro utilizes prediction models derived from multiple databases within the InterPro consortium to classify proteins according to their distinctive characteristics. The results generated by the Interpro database were validated using SMART (http://smart.embl-heidelberg.de/) [19], a tool that extensively annotates a protein domain based on functional class, phyletic distributions, tertiary structures, and functionally important residues. We further validated the results using PROSITE

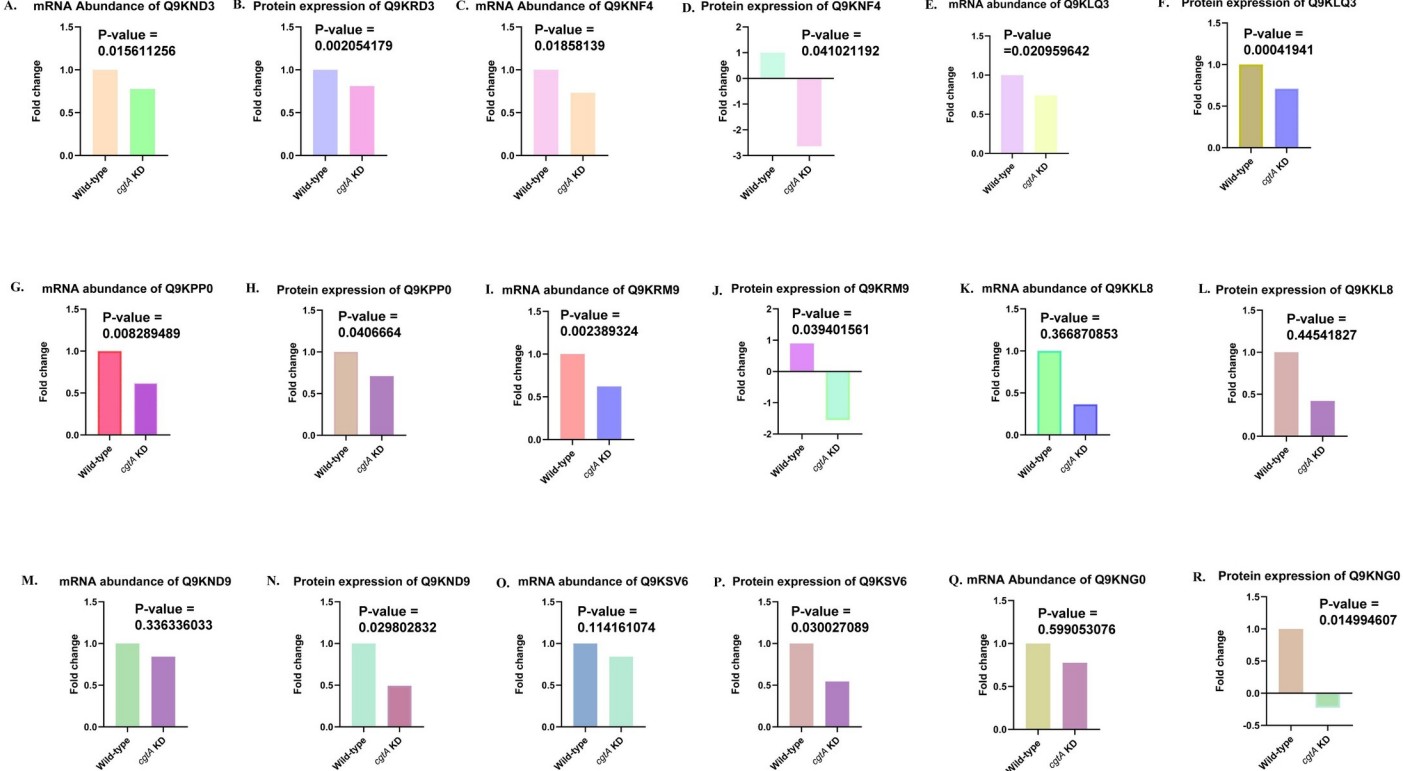

**Fig 2.** The alteration in the expression pattern of transcripts and proteins in CgtA-depleted condition: **A.** mRNA abundance of VC_A0032 (UniProt ID- Q9KND3) shows significant downregulation in the expression of transcript in *cgtA* knockdown strain of *Vibrio cholerae*. **B.** Protein expression of Q9KND3 was significantly reduced when *cgtA* was knocked down from the genome of *V. cholerae* **C.** The mRNA abundance of VC_A0010 (UniProt ID- Q9KNF4) depicts significant downregulation in CgtA depleted condition in *V. cholerae*. **D.** Protein expression levels of Q9KNF4 were downregulated in *cgtA* knockdown strain of *V. cholerae* **E.** The expression of mRNA from VC_0689 (UniProt ID- Q9KLQ3) was significantly reduced in *cgtA* knockdown strain of *V. cholerae*. **F.** The expression of protein, Q9KLQ3 was observed to be significantly reduced in CgtA-depleted condition in *V. cholerae*. **G.** mRNA production from VC_2326 (UniProt ID- Q9KPP0) was significantly downregulated in *cgtA* knockdown strain of *V. cholerae* **H.** Protein expression of Q9KPP0 was significantly reduced in *cgtA* knockdown strain of *V. cholerae* **I.** mRNA production from the gene VC_1607 (UniProt ID-Q9KRM9) was significantly reduced in *cgtA* knockdown strain of *V. cholerae* and **J.** Protein expression of Q9KRM9 was observed to be reduced in CgtA-depleted condition of *V. cholerae* **K.** mRNA production of VC_A0185 (UniProt ID- Q9KKL8) and **L.** protein expression of Q9KKL8 were significantly downregulated in *cgtA* knockdown strain of *V. cholerae* **M.** mRNA expression from VC_A0026 (Q9KND9) and **N.** protein expression of Q9KND9 were downregulated in CgtA depleted condition in *V. cholerae* **O.** mRNA production from VC_1150 (UniProt ID-Q9KSV6) and **P.** protein expression of Q9KSV6 were downregulated in CgtA depleted condition in *V. cholerae* **Q.** mRNA expression from VC_A0004 (UniProt ID- Q9KNG0) and **R.** protein expression of Q9KNG0 were downregulated in *cgtA* knockdown strain of *V. cholerae*.

(https://prosite.expasy.org/) [20], a large database of protein domains, families, and functional sites that identifies conserved sequences and functional site within candidate proteins, which is crucial for understanding their structure and function (**S5 Table**).

## Assessment of hydrophobic and hydrophilic regions

The solubility of uncharacterized proteins was assessed using SOSUI (https://harrier. nagahama-i-bio.ac.jp/sosui/) [21], DeepTMHMM (https://dtu.biolib.com/DeepTMHMM) [22], and HMMTOP (http://www.enzim.hu/hmmtop/) [23]. The SOSUI system could differentiate membrane proteins from soluble proteins and predict transmembrane helices with 99% and 97% accuracy, respectively. Additionally, DeepTMHMM, a deep learning protein language model-based system, accurately detects and predicts alpha-helical and beta-barrel protein structures. We also employed HMMTOP to validate our results by using the hidden Markov model to predict transmembrane protein structure and topological information based on amino acid composition. Additionally, ExPASy's ProtScale (https://web.expasy.org/protscale/)

was utilized to calculate Kyte-Doolittle Hydropathy Plots [24]. The plots revealed the hydrophobic and hydrophilic areas of 51 uncharacterized proteins (**S1 Fig**). The plots allowed us to determine the hydrophobic and hydrophilic regions within the three-dimensional structure of 51 uncharacterized and hypothetical proteins. Plots above 0 (zero) in the graph indicate the hydrophobic region and plots below 0 (zero) indicate the hydrophilic regions within a protein. The results are summarized in **S6 Table.**

## Determination of putative biological processes and molecular functions of candidate proteins

The potential molecular functions, biological processes, and cellular compartments associated with the 51 uncharacterized proteins were analyzed using the PFP (Protein Function Prediction) tool (http://kiharalab.org/web/pfp.php) [25] (**S7 Table**). The protein function prediction tool is an automated method that uses the extended similarity group (ESG) algorithm to forecast the potential biological and molecular activities of proteins whose cellular roles are unknown. Further, "Argot2" (Annotation Retrieval of Gene Ontology Terms) (http://www.medcomp.medicina.unipd.it/Argot2/) [26] was used to validate the results of PFP, which generates output in the form of Gene Ontology (GO) annotations (**S8 Table**).

## Protein–protein interaction prediction

Using the Search Tool for the Retrieval of Interacting Genes/Proteins (STRING version 11.5) (http://string-db.org/), the protein interactions were predicted. This database contains predictions for protein interactions with 14094 species and 67.6 million proteins. The genetic background, curated databases, high-throughput studies, conserved expression, and prior knowledge were the sources of the connections, which comprised both functional and physical relationships [27]. Results were shown with protein scores greater than 0.444 (**S9 Table** **and S2 Fig**).

## Prediction of secondary structures and disordered regions

PSIPRED Protein Analysis Workbench (http://bioinf.cs.ucl.ac.uk/psipred/) is a tool used for determining and predicting the secondary structure of uncharacterized proteins [28]. It includes the ability to connect to GenTHREADER for protein fold identification and MEMSAT-2 for transmembrane topology prediction. The secondary structures of 51 uncharacterized proteins are shown in **S5 Fig**. Further, DISOPRED Workbench, available at http://bioinf.cs.ucl.ac.uk/psipred/, was utilized to identify the disordered area of a protein. The service receives a solitary protein amino acid sequence in FASTA format as input and provides the likelihood of the amino acid in the sequence being disordered as output. The threshold chance for an amino acid residue to create a disordered area is 0.5 (**S4 Fig**).

## Three-dimensional protein structure construction and validation

3-dimensional structures of proteins were constructed with the aid of Alphafold2-mmseqs2 [29]. The model's quality was evaluated through the use of pLDDT and PAE graphs. The 3-dimesnsional model structures of candidate proteins were validated using ProSA (https://prosa.services.came.sbg.ac.at/prosa.php) [30] and PROCHECK (https://www.ebi.ac.uk/thornton-srv/software/PROCHECK/) [31] (**S5 Fig**). The PROCHECK tool evaluates the overall geometry of a model by analyzing the geometry of each residue. It provides insight into the stereochemical quality of the predicted model, indicating whether the residues fall within the most favored regions, additionally allowed regions, generously allowed regions, or disallowed

regions. Overall, a 3D structural model is considered to be of good quality if 90% of the residues are present in the most favored regions.

## Bacterial strains, plasmids, and culture conditions

*V. cholerae* and *Escherichia coli* strains were routinely grown aerobically in Luria–Bertani (LB) medium (10g/L NaCl, 10g/L Tryptone and 5g/L Yeast Extract, pH = 7.0) at 37˚C. The agar medium contained 1.5% (wt/vol) agar, except for the motility studies, where the agar concentration was 0.3% and 0.5% (wt/vol). *V. cholerae* was grown for the genomic DNA isolation required for PCR amplification. The pET28a (+) vector was used for cloning the uncharacterized proteins. The *Escherichia coli* DH5-alpha strain was used for amplifying and screening clones. However, *Escherichia coli* BL21 (pLyss) cells were used for the induction and expression of proteins.

## Cloning, expression, and purification of uncharacterized proteins

Two full-length genes expressing uncharacterized proteins were cloned into the expression vector pET28a (+). The two proteins (a large 92kDa and a small 12kDa protein) were selected based on their ability to be potential vaccine candidates as predicted by *in silico* studies. The forward and reverse primers for amplifying the protein (UniProt ID-Q9KRD2) were TAAGC AGGATCCATGAGTGTGAATGTATCAACCGT and TAAGCACTCGAGTCAACTCGCTAAATA AGCGAGCA, respectively. The forward and reverse primers for amplifying the proteins (Uni-Prot ID- Q9KU58) were TAAGCAGGATCCGTGTCTTCTGACTTTTCCCT and TAAGCACTC GAGTTACGTCGGTATTCGCG, respectively. The restriction sites for BamHI and XhoI were added for the overhang production necessary for cloning purposes. The gene was subsequently cloned and inserted into the pET28a (+) vector in such a way that the N-terminal His tag was added. Protein production was carried out in LB medium via IPTG induction (1 mM final concentration) at 37˚C and 160 rpm. *E. coli* cells harboring amplified gene products were subsequently harvested, resuspended in lysis buffer (20 mM Tris/HCl pH 8, 200 mM NaCl, and 20 mM imidazole) supplemented with 1mM EDTA-free protease inhibitor cocktail, and lysed using a sonicator (pulse on 5 seconds, pulse off on 20 seconds, amplitude 60%). The lysed cells were centrifuged at $11000 \times g$ at 4˚C. The inclusion bodies pellet was washed with washing buffer (1X PBS, 1% Triton-X-100, 1M Urea) and was dissolved and stored in 1X PBS and 2M urea at -20˚C overnight. The solubilized inclusion body was then purified using a Ni-NTA column. The expression of the proteins was further validated by performing western blotting using an anti-His tag antibody.

## Antigenicity, allergenicity, toxicity, and virulence of proteins

The antigenicity properties of 51 uncharacterized proteins were assessed using VaxiJen (http://www.ddg-pharmfac.net/vaxijen/VaxiJen/VaxiJen.html) [32], with a cutoff score of 0.4. Moreover, the allergenicity of each uncharacterized protein was evaluated using the AllerTOP v.2.0 tool (https://www.ddg-pharmfac.net/AllerTOP/) [33]. Additionally, the toxicity of the candidate proteins was determined using the ToxinPred 2 tool (https://webs.iiitd.edu.in/raghava/toxinpred2/batch.html) [34], which employs a hybrid machine learning technique with a cut-off threshold of 0.6 (**S10 Table**). The virulence of a protein was predicted using the VirulentPred (http://bioinfo.icgeb.res.in/virulent/) [35] tool, which utilizes the SVM method to predict protein virulence (**S11 Table**).

## B-cell epitopes prediction

Linear B-cell epitope prediction was performed using BepiPred-3.0 (http://tools. The iedb.org/bcell/) [36]. It utilizes a cutoff value of > 0.5 to predict linear B-cell epitopes based on protein language models (LMs). Subsequently, BcePred (https://webs.iiitd.edu.in/raghava/bcepr ed/bcepred_submission.html) [37], which predicts epitopes based on physicochemical properties like hydrophilicity, polarity and surface properties, was also employed to predict the linear B-cell epitopes (**S12 Table** and **S6 Fig**). The prediction of discontinuous B-cell epitopes for the identified uncharacterized protein was performed using DiscoTope (http://tools.iedb.org/discotope/help/) [38]. The prediction was based on the 3D structures of proteins in PDB format (**S13 Table** and **S7 Fig**).

## Determination of T-cell epitopes

The NetCTL (http://www.cbs.dtu.dk/services/NetCTL/) servers were used to predict the T-cell epitopes of the proteins. The NetCTL server was used to predict cytotoxic T-lymphocyte (CTL) epitopes of the query proteins [33]. NetCTL prediction depends on the binding affinity of MHCI, C-terminal cleavage function, and transporter function associated with antigen processing. The combined threshold value for MHC-I prediction is 0.75. The retrieved information is tabulated in **S14 Table**. The IEDB MHCII (http://tools.iedb.org/mhcii/) server was used for identifying helper T-lymphocyte (HTL) epitopes, as shown in **S15 Table**. Human/HLA-DR was chosen as the species/locus with 7 alleles of human leukocyte antigen (HLA), and 15-m-long epitopes were retrieved.

# Results and discussion

## Transcriptomics and proteomics data reveals uncharacterized proteins in *ΔcgtA V. cholerae*

Transcriptomic analysis via RNA-seq and a label-free proteomics study revealed numerous transcripts and proteins whose expression were altered in *V. cholerae* under CgtA-depleted conditions. CgtA is an essential ribosome-associated GTPase that has multiple functions. On knocking down *cgtA* from cholerae causing *Vibrio cholerae*, around 51 proteins have been identified whose role are yet to be deciphered. Among all the 51 uncharacterized proteins, fifty proteins were downregulated at mRNA transcript level, 9 proteins were downregulated at both mRNA and protein level and the protein Q9KU58 was upregulated in *V. cholerae* N16961 *ΔcgtA*: *kanR*/*cgtA*-pBAD18Cm. The fold changes and p values are noted in **S1 Table**. **Table 1** shows the comparison of the proteomic and transcriptomic data for 9 proteins- Q9KND3, Q9KNF4, Q9KLQ3, Q9KPP0, Q9KRM9, Q9KKL8, Q9KND9, Q9KSV6, and Q9KNG0 whose expression was altered at mRNA and protein level, when *cgtA* was knocked down. **Fig 2A–2I** show the genes and their products that were downregulated in *V. cholerae* N16961 *ΔcgtA*: *kanR*/*cgtA*-pBAD18Cm with respect to the wild-type strain.

## Physicochemical properties of the uncharacterized proteins

The physicochemical properties of the fifty-one uncharacterized and hypothetical proteins, including the size of the protein, molecular weight, theoretical isoelectric point (pI), molar extinction coefficient, instability index, aliphatic index, grand average hydropathy (GRAVY), and the total number of positively charged (Arg+Lys) and negatively charged (Asp+Glu) residues, are listed in **S2 Table**. The proteins are arranged in descending order of length. The amino acid composition found to include all 20 standard amino acids, each present at varying percentages (**S3 Table**). The top 10 amino acids, in order of abundance, were D, T, I, Q, K, V,

**Table 1. Comparison between label-free proteomics and transcriptomics analyses.**

| Gene and Protein Identifier | | Protein Expression | | mRNA abundance | |
| --- | --- | --- | --- | --- | --- |
| Gene ID | Protein ID | Log₂Fold change | p value | Log₂Fold change | p value |
| Q9KND3 | VC_A0032 | -2.1631343 | 0.002054179 | -2.03968639 | 0.015611256 |
| Q9KNF4 | VC_A0010 | 1.877225002 | 0.041021192 | -1.89614697 | 0.01858139 |
| Q9KLQ3 | VC_A0689 | -1.943823 | 0.00041941 | -1.78636172 | 0.020959642 |
| Q9KPP0 | VC_2326 | -1.78433722 | 0.008289489 | -1.40274321 | 0.0406664 |
| Q9KRM9 | VC_1607 | 1.352318913 | 0.002389324 | -1.94337726 | 0.039401561 |
| Q9KKL8 | VC_A0185 | -0.783152317 | 0.366870853 | -.654626335 | 0.044541827 |
| Q9KND9 | VC_A0026 | -0.979426671 | 0.336336033 | -2.662460158 | 0.029802832 |
| Q9KSV6 | VC_1150 | -1.134660702 | 0.114161074 | -2.662277492 | 0.030027089 |
| Q9KNG0 | VC_A0004 | 0.292646614 | 0.599053076 | -2.15772107 | 0.014994607 |

Out of 51 proteins, there are 9 proteins whose altered expressions were detected in both the RNA-seq and label free proteomic data. The log₂Fold change and p-value indicating the alteration in expression of transcript and proteins are recorded for each protein.

S, E, A, and L (**Fig 3A**). Additionally, the uncharacterized proteins exhibited a molecular weight range of 5.19 kDa to 91.831 kDa, and a pI fall of 4.19 to 11.40 (**S2 Table** and **Fig 3B and 3C**). Moreover, the uncharacterized proteins were exhibited a range of lengths, spanning from 45 to 818. Additionally, the aliphatic index of the protein ranged from 10.62 to 135.36, as indicated in **S2 Table**. In addition, the protein exhibited a negative GRAVY score ranging from -0.967 to -0.0268 (**S2 Table** and **Fig 3D**). The proteins exhibited an instability index ranging from 17.6 to 101.35, with stable proteins having an optimal score of less than 40 on the instability index. Among the 51 proteins, only 29 of them were found to be stable in their natural state, as indicated in **S2 Table**. Furthermore, among the 51 proteins, a total of 10 proteins (Uniprot ID-Q9KKX0, Q9KVJ9, Q9KP29, Q9KPA3, Q9KT53, Q9KL56, Q9KRE6, Q9K2J6,

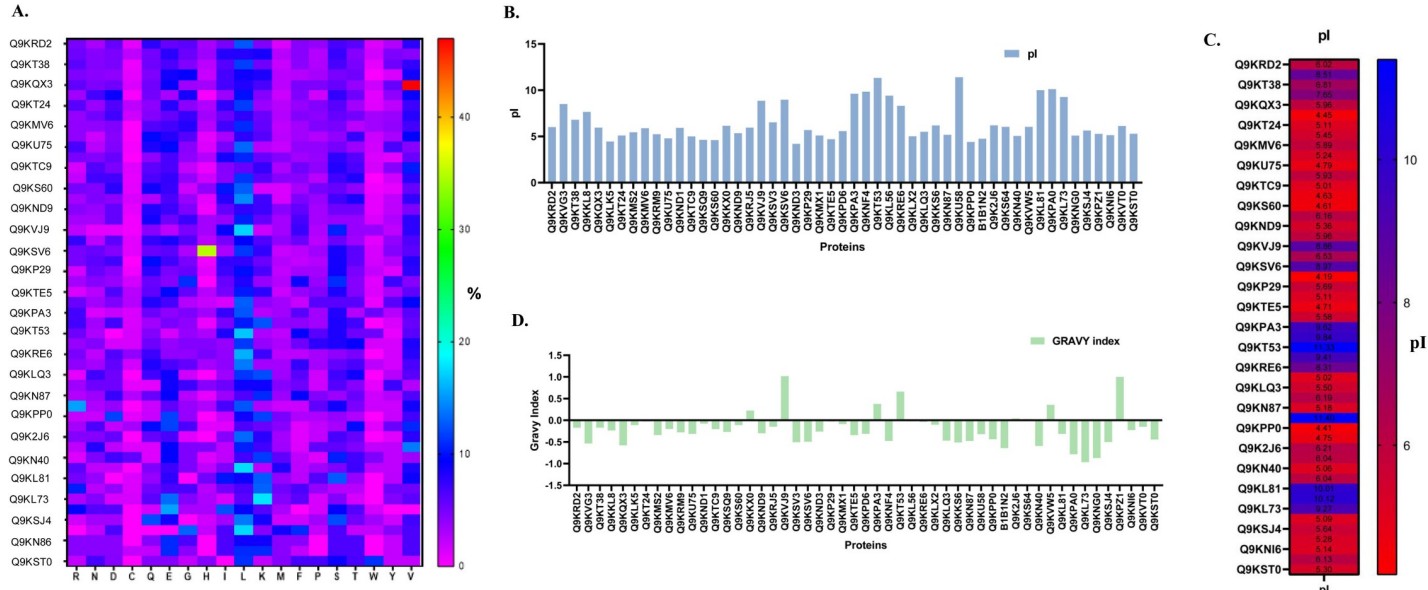

**Fig 3.** Physicochemical properties of 51 uncharacterized proteins: A. Heatmap showing the amino acid composition of each protein. The scale on the right is expressed in terms of percentage. **B.** The bar graph illustrates the pI of all the uncharacterized proteins. Majority of the proteins were seen to be acidic **C.** The heatmap shows the pI of all the uncharacterized proteins, where the scale represents the pH. **D.** The GRAVY index indicates the solubility of all the fifty-one candidate proteins. A negative GRAVY value indicates that a protein is hydrophilic in nature. Majority of the uncharacterized proteins were predicted to be soluble.

Q9KVW5, and Q9KPZ1) have been determined to be insoluble in their natural state. A total of 15 positively charged proteins (rich in Arginine and Lysine) and 24 negatively charged proteins (rich in Aspartic Acid and Glutamic Acid) were found out of the 51 uncharacterized proteins (S2 Table).

### Predicted subcellular localization of the uncharacterized proteins

The subcellular localization of the 51 uncharacterized proteins was determined using two tools: PSLPred, and Psortb. The results indicate that the proteins are abundant in the cytoplasmic, cytoplasmic membrane, and outer membrane compartments, as determined by the PSORTb score (7.88–10) (S4 Table and Fig 4A). Additionally, the PSLpred analysis indicate that the proteins are present in the cytoplasmic, outer-membrane, periplasmic, inner-membrane, and extracellular compartments, with an expected accuracy ranging from 53.1% to 98.10% (S4 Table and Fig 4B). Additionally, the PFP and Argot2 tools indicated that the uncharacterized proteins are found throughout the cell, namely in the membrane and cytoplasm (Fig 4C and 4D).

### Solubility and transmembrane helices identification uncharacterized proteins

An analysis has been conducted on the solubility, protein type, and transmembrane helix region of 51 uncharacterized proteins using three machine learning tools: SOSUI, DeepTMHMM, and HMMTOP. The SOSUI analysis revealed that 11 proteins possess

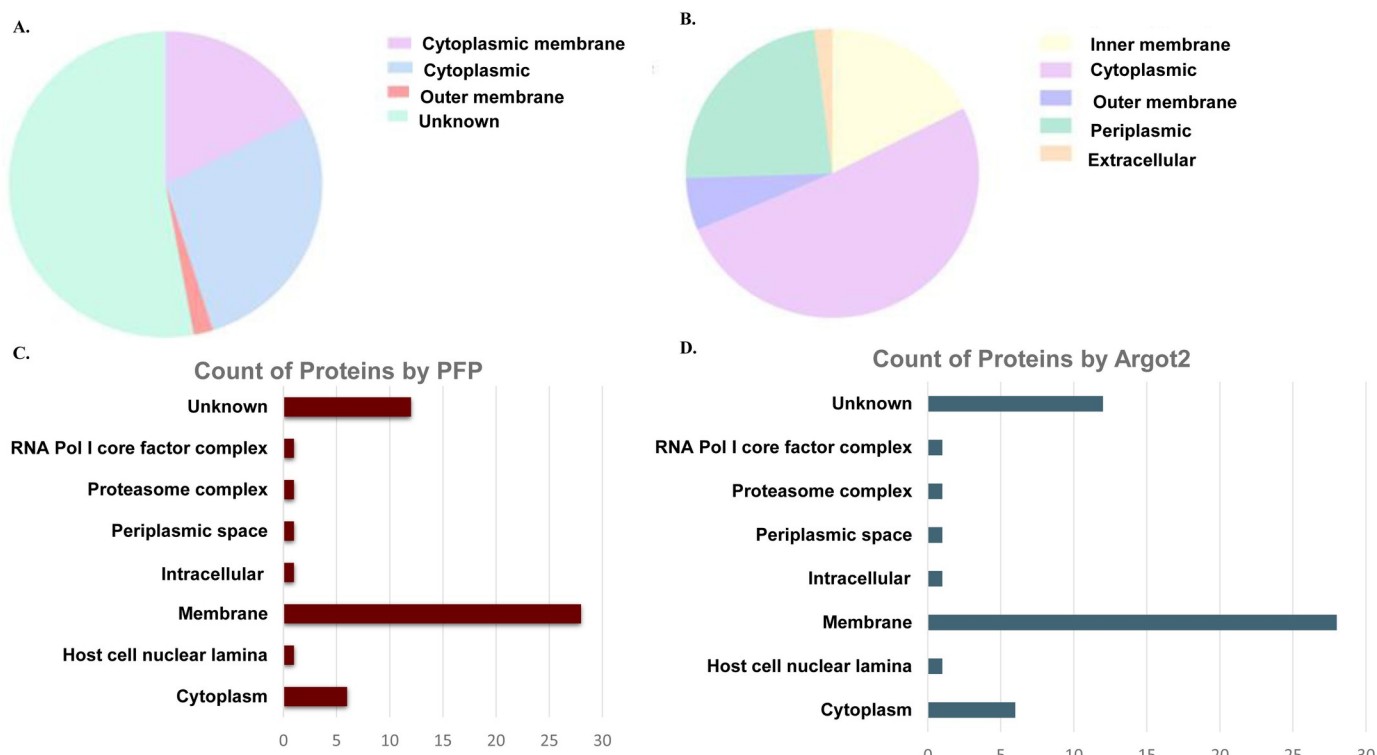

**Fig 4.** Prediction of subcellular localization of 51 uncharacterized proteins: **A.** A pie chart illustrating the subcellular location of uncharacterized and hypothetical proteins predicted by Psortb. **B.** A pie chart depicting the subcellular location of uncharacterized and hypothetical proteins predicted by PSLPred. **C.** Predicted subcellular location of uncharacterized and hypothetical proteins predicted by PFP (Protein Function Prediction) and **D.** Argot2. Majority of the proteins were found to be associated with cytoplasm and membrane.

transmembrane helices and are classified as membrane proteins, while 40 proteins are classified as globular and soluble in nature (**S6 Table**). In addition, the hydropathy plot generated by ExPASy's ProtScale tool displays the profile proteins based on position and amino acid composition hydrophobicity or hydrophilicity are displayed in the scale of -10 to +10 (**S1 Fig**).

### Molecular function and biological process of the uncharacterized proteins

The molecular function of each uncharacterized protein was confirmed using the PFP (**Fig 5A**) and Argot2 tools (**Fig 5B**). The results indicated that the majority of the proteins were associated with ATP binding, DNA binding, RNA binding, translation elongation factor, and motor activity. This was determined based on the PFP Score range of 1.22 to 14120.19 (**S7 Table**) and the Argot2 score range of 5.158 to 11459.4 (**S8 Table**). In accordance to the biological process analysis (Argot2), a large number of proteins have been linked in the regulation of transcription, DNA integration, cell morphogenesis, chemotaxis, and metabolism. The finding is based on the Argot2 score range of 6.26808 to 38638.5 (**Fig 5C and 5D**). A few uncharacterized proteins have also been shown to play a crucial role in motility, aerotaxis, and chemotaxis. A comparison was carried out to examine the swimming and swarming motility of the CgtA-depleted strain and the wild-type strain of *V. cholera*. The findings indicate that in the *cgtA* knockdown strain, the swimming motility was significantly reduced compared to the wild-type strain of *V. cholera* (**Fig 5D**). The diameter of the swimming motility zone was determined and statistically verified using a two-tailed t-test at a significance level of $p < 0.05$ (**Fig 5E**). Similarly, the swarming motility of the *cgtA* knockdown strain was found to be considerably lower than that of the wild-type strain of *V. cholera* (**Fig 5F and 5G**). However, the mechanism bridging the effect of motility and CgtA is yet to be determined. We can strongly anticipate and explore the mechanism of bacterial motility and pathogenesis through further metabolomic studies and functionally characterizing the uncharacterized proteins. Hence, to summarize, understanding the role of these proteins will enable us to look at the bigger picture of cholera pathogenesis, which affects millions of people globally.

### Identification of protein-protein interaction network

Protein-protein interaction or PPI network for 51 uncharacterized proteins were visualized in STRING version 11.5, which predicts the interactive protein partners using a confidence score above 0.7. The interacting partners of each of the uncharacterized proteins are tabulated in **S15 Table**. The highest confidence score of 0.996 and 0.994 were observed for the interaction between SpoVR family protein, Q9KQX3 and a hypothetical protein, Q9KQX4 (VC_1873) and uncharacterized protein, Q9KQX5 (VC_1872) respectively. Other interaction with high confidence score of 0.993 was seen in PPI between EAL domain containing Q9KRD2 and Q9KPJ7 (Gene name- VC_2370) which is a sensory box containing diguanylate cyclase enzyme. The findings in this study will allow us to reinforce our understanding on the probable functions of these uncharacterized proteins and decipher the molecular pathways involved with cellular functions.

### Three-dimensional protein structure construction and validation

The 3-dimensional structure of 51 uncharacterized proteins were predicted by AalphaFold-mmseqs2 (**S5 Fig**). Only the best ranked structure is shown which is validated by predicted local distance difference test (pLDDT) score and predicted aligned error (PAE) graph. pLDDT value above 90 indicates high model confidence and accurate structure. A protein with Uni-Prot ID- Q9KMS2, a 40.5 kDa was observed to have the highest model confidence with

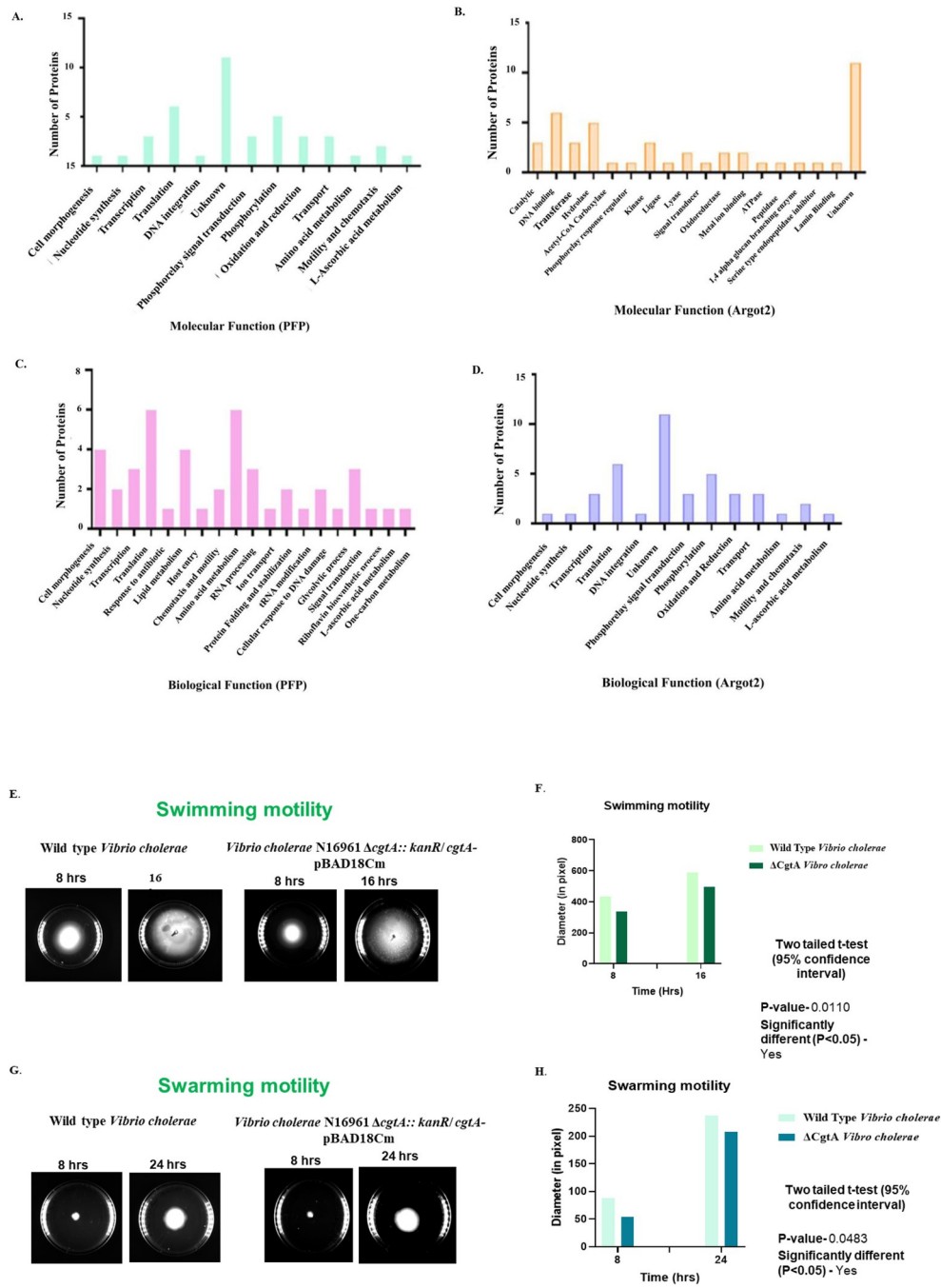

**Fig 5.** Prediction of functions of 51 uncharacterized proteins **A.** The green-colored bar chart illustrates the probable molecular functions associated with the 51 uncharacterized and hypothetical proteins predicted by PFP (Protein Function Prediction) tool. Majority of the proteins were found to involved in essential cellular functions like transcription, translation, phosphorylation, transport, chemotaxis and motility. **B.** The yellow-colored bar chart depicts the predicted molecular function of uncharacterized and hypothetical proteins predicted by Argot2. According to Argot2 prediction, majority of the proteins were involved with catalytic activity and DNA binding. **C.** The pink-colored bar graph illustrates the probable biological functions of uncharacterized and hypothetical proteins predicted by PFP. Majority of the proteins were found to be involved with cellular morphogenesis, translation, ribosome biogenesis and amino acid metabolism **D.** The blue-colored bar graph indicates the probable biological functions of uncharacterized and hypothetical proteins predicted by Argot2, where majority of the proteins were found to be involved with transcription, translation, signal transduction, phosphorylation and transport **E.** Analysis and comparison of swimming motility between the CgtA-depleted strain and the wild-type strain of *V. cholerae*. In the *cgtA* knockdown strain, the swimming motility were significantly lower than those in the wild-type strain of *V. cholerae*.

The diameter of motility zone was measured and **F.** statistically validated using a two-tailed t test. **G.** Analysis and comparison of swarming motility between the CgtA-depleted strain and the wild-type strain of *V. cholerae*. Similar to swimming motility, the swarming motility was also significantly lower than those in the wild-type strain of *V. cholerae*. The diameter of motility zone was measured and **H.** statistically validated using a two-tailed t test.

pLDDT score of 97.9. Further, the structures were validated using PROCHECK Ramachandran Plot and ProSA.

## Cloning, expression and purification of uncharacterized proteins

To validate the results derived from the *in-silico* analysis, we have selected two proteins (a 92 kDa Q9KRD2 and 12 kDa Q9KU58 as predicted in ProtParam) to perform cloning and purification since they approximately cover the entire size range, i.e., 92 kDa of the set of the uncharacterized proteins used for this study. In addition, RNA-seq data showed that the expression of the large protein with UniProt Id-Q9KRD2 was downregulated (**Fig 6A**), and the label-free proteomic data revealed that the small protein with UniProt Id-Q9KU58 was significantly upregulated when *cgtA* was knocked down in *Vbrio cholerae* (**Fig 6B**). We isolated genomic DNA from *V. cholerae* (**Fig 6C**) and amplified two genes of interest by quantitative polymerase chain reaction (**Fig 6D and 6E**). We digested and cloned the amplified product (product size Q9KRD2: 2457 bp, and Q9KU58:315 bp) into the pET28a(+) vector and transformed it into chemically competent *Escherichia coli* DH5α. The clones (**Fig 6F**) were screened and transformed into chemically competent *Escherichia coli* Bl21 (pLySs) cells. Further, cells were then cultivated in Luria–Bertani liquid media up to an O.D. of 0.7 and then induced with 1mM IPTG and incubated at 37°C for 5 hours. The cells were then induced with 1mM IPTG and incubated at 37°C for 5 hours. The induced cells were then sonicated (60% amplitude, pulse on = 5 seconds, pulse off = 15 seconds) for 15 minutes, and the cell lysates were centrifuged at 11000 × g for 10 minutes at 4°C. The supernatant was kept aside and the inclusion body pellet was solubilized in PBS buffer and 2M urea. The GRAVY index for Q9KRD2 and Q9KU58 were predicted to be -0.172 and -0.319, respectively, which indicates their solubility in solution. Also, the predictions based on SOSUI and DeepTMHMM, the two proteins were predicted to be soluble. However, while expressing the proteins in the heterologous system of *E. coli* BL21 (pLySs) both the proteins showed solubility issues. We have used mild concentration (2M) of polar reagent, urea, to solubilize the inclusion body. The solubilized inclusion body were then purified using a nickel-NTA column. Finally, the expression of the purified protein Q9KRD2 (**Fig 6G**) and Q9KU58 (**Fig 6I**) were checked by SDS–PAGE, respectively. The purified Q9KRD2 showed a clear band near 100 kDa protein marker and purified Q9KU58 showed a distinct band near 15 kDa band as expected from *in silico* analysis. Further, the expression of the two purified protein were validated western blot analysis (**Fig 6H and 6J**).

The 3-dimensional model structure of Q9KRD2 and Q9KU58 is shown in (**Fig 6K and 6P**). Ramachandran plot validation for the protein Q9KRD2 suggested that 91.70% the residues were seen to be in the most favored region (**Fig 6L and 6M**) while for Q9KU58 79.30% (**Fig 6Q and 6R**). Additionally, ProSA z-score of Q9KRD2 and Q9KU58 found as -13.31 (**Fig 6N**) and -2.66 (**Fig 6S**) respectively ProSA was also used for assessing local model quality of Q9KRD2 (**Fig 6O**) and Q9KU58 (**Fig 6T**).

## Determination of the antigenicity, allergenicity, toxicity, and virulence of proteins

Understanding the pathogenesis of a disease is extremely vital for designing drugs to combat and alleviate the occurrence of deadly diseases such as cholera. To obtain a larger and clearer

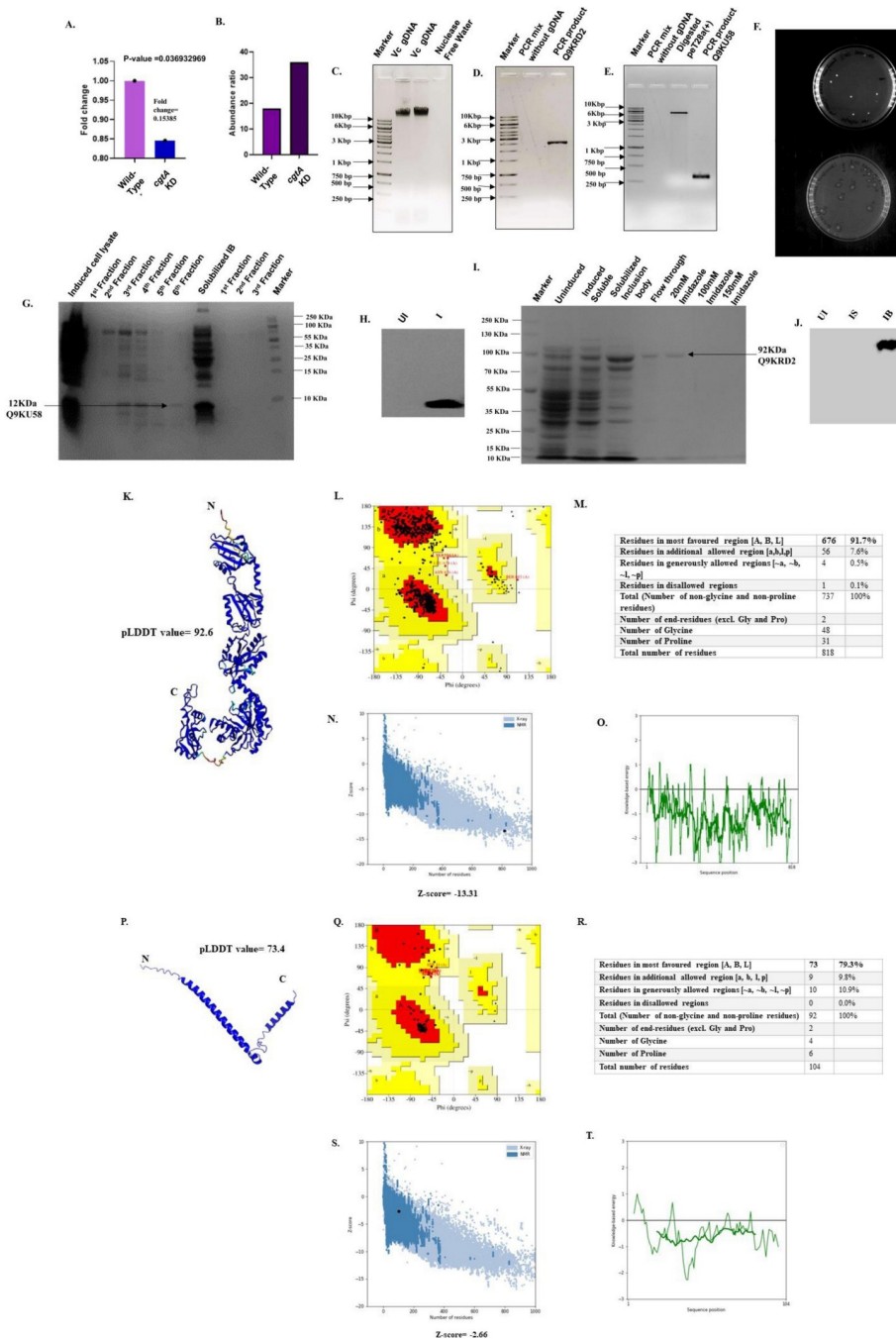

**Fig 6.** Cloning, expression and purification of two uncharacterized proteins: A. Fold change in mRNA expression shows that the EAL domain-containing Q9KRD2 is downregulated in CgtA-depleted cells. **B.** The abundance ratio shows that the expression of the protein Q9KU58 is upregulated in CgtA-depleted conditions. **C.** Genomic DNA isolated from wild-type *V. cholerae*. Lane 1 = marker, Lane 2 and 3 = isolated genomic DNA from *V. cholerae*, Lane 3 = Nuclease free water (Control) **D.** PCR amplification of Q9KRD2. Lane 1 = marker. Lane 2 = PCR mix without genomic DNA (control), Lane 3 = amplified product (length = 2547bp). **E.** PCR amplification of Q9KU58. Lane 1 = marker. Lane 2 = PCR mix without genomic DNA (control), Lane 3 = digested peT28a (+). Lane 4 = amplified product (315 bp). **F.** Luria Bertini agar plates showing *Escherichia coli* DH5α colonies containing chimeric plasmid containing gene expressing for Q9KRD2 (top) and Q9KU58 (bottom). **G.** Expression of 12 kDa protein, Q9KU58 in lane 7 (purified soluble fraction). **H.** Western blot gel verifying the expression of Q9KU58. Lane 1 = Uninduced, Lane 2 = induced. **I.** Expression of 92kDa protein, Q9KRD2 in lane 5 (purified solubilized inclusion body fraction). **J.** Western blot analysis verifying the expression of Q9KRD2. Lane 1 = Uninduced, Lane 2 = induced solubilized fraction, Lane 3 = induced solubilized inclusion body fraction. **K.**3-dimensional structure of the protein with UniProt ID–

Q9KRD2, predicted by AlphaFold-mmseqs2 **L.** Ramachandran plot for 3-dimensional structure of protein with UniProt ID- Q9KRD2. **M.** Plot statistics of Ramachandran plot for protein with UniProt ID- Q9KRD2. **N.** Plot depicting the overall model quality of the protein with UniProt ID- Q9KRD2. The Z-score, which indicates the overall model quality, is -13.31. **O.** Plot depicting the local model quality of protein with UniProT ID- Q9KRD2 **P.** 3-dimensional structure of the protein with UniProt ID–Q9KU58, predicted by AlphaFold-mmseqs2 **Q.** Ramachandran plot for the 3-dimensional structure of the protein with UniProt ID- Q9KU58. **R.** Plot statistics of Ramachandran plot for protein with UniProt ID- Q9KU58. **S.** Plot depicting the overall model quality of the protein with UniProt ID- Q9KU58. The Z-score, which indicates the overall model quality, is -2.66. **T.** Plot depicting the local model quality of protein with UniProT ID- Q9KU58.

picture of cholera pathogenesis, characterizing and understanding the roles of proteins that are associated with the main switch, i.e., CgtA, an essential ribosome-associated GTPase that plays multifarious cellular roles required for the survival of bacterial cells, are essential. The solubility, antigenicity, allergenicity, toxicity, and virulence of the proteins were assessed, as shown in **Fig 6**. We identified potential vaccine candidates and drug targets from the pool of uncharacterized proteins by a comprehensive reverse vaccinology study, as shown in **S10–S15 Tables.** The antigenicity profiles were determined using the VaxiJen, which exploits alignment-independent prediction of protective antigens. The VaxiJen results were analyzed based on the VaxiJen Overall prediction score, where scores greater than 4 indicate antigenic proteins and scores less than 4 indicate non-antigenic proteins. The results showed that out of the 51 uncharacterized proteins, 31 were found to be potent antigens (with VaxiJen Overall prediction scores ranging from 0.4098 to 0.5438), while 20 were non-antigenic (with VaxiJen Overall prediction scores ranging from 0.2393 to 0.3949), meaning they do not elicit an immune response (**S10 Table** and **Fig 7A and 7D**). The AllerTop tool result indicated that out of 51 proteins, 43 were classified as non-allergens, while the remaining proteins were identified as probable allergens (**S10 Table** and **Fig 7B**). In addition, the toxicity profiles of the candidate's proteins were determined using the hybrid score obtained from the ToxinPred 2 program. The ToxinPred analysis generated a hybrid score indicating that 49 proteins were non-toxic, while only 2 proteins were toxic (**Fig 7C**). The hybrid score for the non-toxic proteins varied from -0.32 to 0.49, while for the toxic proteins, it ranged from 0.68 to 0.79 (**S10 Table**). On screening the proteins based on antigenicity, allergenicity, toxicity, B-cell (**S12** and **S13 Tables**) and T-cell epitope prediction (**S14** and **S15 Tables**), only 20 proteins were found to be potential vaccine candidates, as shown in **Table 2,** which can be further validated by experimental studies. We also found several potential drug targets, as determined by their ability to be virulent proteins, as depicted in **S11 Table.** The virulence of the proteins was assessed using the VirulentPred program, which provided findings based on five separate parameters: amino acid composition, dipeptide composition, PSI-BLAST generated PSSM profiles, a cascade of SVMs and PSI-BLAST, and higher order dipeptide composition. In the amino acid composition-based method, there were a total of 34 proteins that were classified as virulent, while 17 proteins were classified as non-virulent. The scores for the non-virulent proteins ranged from -1.873 to -0.053, while the scores for the virulent proteins varied from 0.0028 to 1.6919. Using the dipeptide composition approach, it was found that out of the 51 proteins analyzed, 22 were non-virulent and 29 were virulent. The virulent proteins had scores ranging from -0.518 to -0.087, while the non-virulent proteins had scores ranging from 0.1159 to 0.4899. Furthermore, PSI-BLAST analysis produced Position Specific Scoring Matrix (PSSM) profiles, which indicated that there were 21 proteins with scores ranging from -1.39 to -0.016, suggesting they were non-virulent. Additionally, there were 30 proteins with scores ranging from 0.0093 to 1.1672, indicating they were virulent. The cascade of Support Vector Machines (SVMs) and PSI-BLAST analysis yielded virulent scores ranging from 0.0182 to 1.1188, and non-virulent scores ranging from -1.1672 to -0.288. In addition to these, 17 proteins were classified as non-

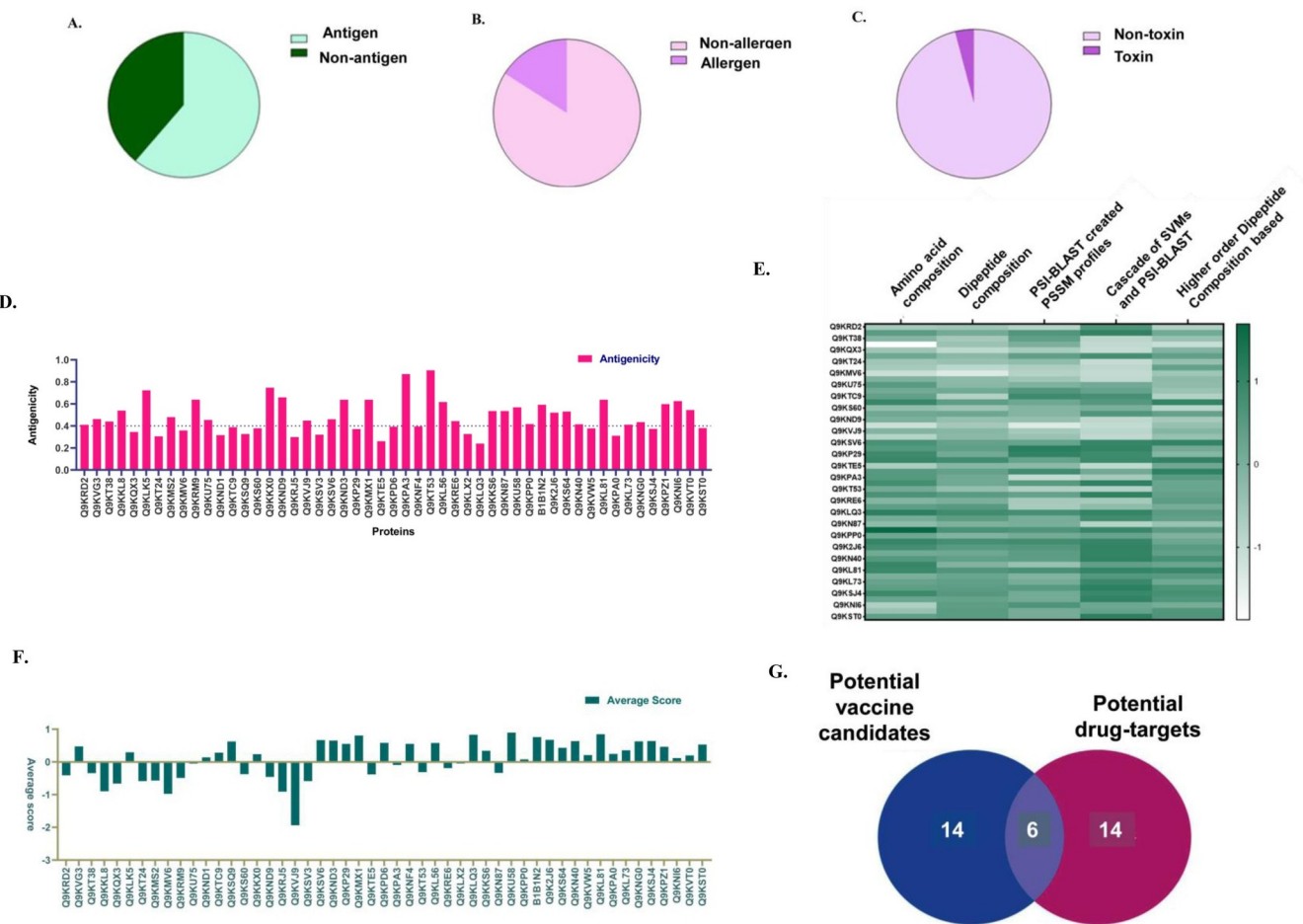

**Fig 7.** Reverse vaccinology and immunoinformatic analysis on 51 uncharacterized proteins **A.** Pie chart illustrating the fraction of proteins that are antigen (Number of antigenic proteins = 31) and non-antigen (Number of non-antigenic protein = 20) out of 51 uncharacterized proteins. **B.** Pie chart depicting the relative fraction of proteins that are allergenic (Number of allergenic proteins = 8) and non-allergenic (Number of non-allergenic protein = 43). **C.** Pie chart illustrating the fraction of uncharacterized proteins that are toxic (Number of toxic proteins = 2) and non-toxic (Number of non-toxic proteins = 49) to human cells. **D.** The bar graph shows that individual proteins that have antigenic properties (cutoff = 0.4) that can elicit an immune response in host cells The X-axis depicts the 51 uncharacterized proteins and Y-axis depicts the antigenic score generated by VaxiJen. **E.** The bar graph illustrates the proteins that are virulent and potential drug targets (threshold = 0), based on the prediction by VirulentPred. The X-axis depicts the 51 uncharacterized proteins and Y-Axis depicts the average hybrid score generated by VirulentPred. **F.** Heatmap showing the virulence of individual proteins predicted by 5 approaches of VirulentPred (amino acid composition, Dipeptide composition, PSI-BLAST created PSSM profiles, cascade of SVMs and PSI-BLAST and higher order dipeptide composition). The scale on the right side depicts the score generated by VirulentPred. **G.** Venn diagram showing the numbers of uncharacterized proteins that have been predicted to be potential vaccine candidates and potential drug targets. Around 20 proteins, were found to be potential drug-targets and 20 proteins were predicted to be potential vaccine candidates. There were 6 proteins that were predicted to be both potential vaccine candidate and drug-target.

virulent and 34 proteins were classified as virulent. An analysis using higher-order dipeptide composition indicated that 18 proteins had a non-virulent nature, with scores ranging from -1.107 to -0.066. In contrast, 33 proteins were found to be virulent, with scores ranging from 0.076 to 1.0795 (**S11 Table** and **Fig 7E**). We further screened the proteins based on the average score of virulence and plotted them as illustrated in **Fig 7F**. On screening, we found around 20 proteins that are potential drug targets as shown in **Table 2** and **Fig 7G**.

## Linear and discontinuous B-cell epitope prediction

Among the 51 uncharacterized proteins, only 24 proteins were investigated to identify the linear and discontinuous B-cell epitopes. Proteins were selected for their high antigenicity, non-

**Table 2. Summary of the proteins that are potential vaccine candidates and drug targets.**

| Potential vaccine candidates predicted by Vaxijen | Potential vaccine candidates predicted by B-cell epitope determination | Potential vaccine candidates predicted by T-cell epitope determination | Potential vaccine candidates which are unanimously predicted by Vaxijen, B-cell a T-cell epitope determination | Potential drug targets predicted by VirulentPred (based on average score) | Potential drug targets that are unanimously virulent in the algorithms of ViruelntPred |
|---|---|---|---|---|---|
| Q9KRD2 | Q9KRD2 | Q9KRD2 | Q9KRD2 | Q9KVG3 | Q9KVG3 |
| Q9KVG3 | Q9KVG3 | Q9KVG3 | Q9KVG3 | Q9KLK5 | Q9KSQ9 |
| Q9KT38 | Q9KT38 | Q9KT38 | Q9KT38 | Q9KU75 | Q9KSV6 |
| Q9KKL8 | Q9KKL8 | Q9KKL8 | Q9KKL8 | Q9KND1 | Q9KND3 |
| Q9KLK5 | Q9KLK5 | Q9KLK5 | Q9KLK5 | Q9KTC9 | Q9KMX1 |
| Q9KU75 | Q9KU75 | Q9KU75 | Q9KU75 | Q9KSQ9 | Q9KPD6 |
| Q9KND9 | Q9KND9 | Q9KND9 | Q9KND9 | Q9KKX0 | Q9KNF4 |
| Q9KVJ9 | Q9KVJ9 | Q9KVJ9 | Q9KVJ9 | Q9KSV6 | Q9KL56 |
| Q9KVJ9 | Q9KSV6 | Q9KSV6 | Q9KSV6 | Q9KND3 | Q9KLQ3 |
| Q9KSV6 | Q9KND3 | Q9KND3 | Q9KND3 | Q9KP29 | Q9KKS6 |
| Q9KND3 | Q9KPA3 | Q9KPA3 | Q9KPA3 | Q9KMX1 | Q9KU58 |
| Q9KPA3 | Q9KT53 | Q9KT53 | Q9KT53 | Q9KPD6 | BIBIN2 |
| Q9KT53 | Q9KRE6 | Q9KRE6 | Q9KRE6 | Q9KNF4 | Q9K2J6 |
| Q9KRE6 | Q9KKS6 | Q9KKS6 | Q9KKS6 | Q9KL56 | Q9KS64 |
| Q9KKS6 | Q9KN87 | Q9KN87 | Q9KN87 | Q9KLX2 | Q9KN40 |
| Q9KN87 | Q9KU58 | Q9KU58 | Q9KU58 | Q9KLQ3 | Q9KL81 |
| Q9KU58 | Q9KPP0 | Q9KPP0 | Q9KPP0 | Q9KKS6 | Q9KL73 |
| Q9KPP0 | BIBIN2 | BIBIN2 | BIBIN2 | Q9KU58 | Q9KNG0 |
| BIBIN2 | Q9KL81 | Q9KL73 | Q9KL73 | Q9KPP0 | Q9KSJ4 |
| Q9KL81 | Q9KL73 | Q9KPZ1 | Q9KPZ1 | B1B1N2 | Q9KPZ1 |
| Q9KL73 | Q9KNG0 | | | Q9K2J6 | |
| Q9KNG0 | Q9KPZ1 | | | Q9KS64 | |
| Q9KPZ1 | Q9KNI6 | | | Q9KN40 | |
| Q9K9I6 | Q9KVT0 | | | Q9KVW5 | |
| Q9KVT0 | | | | Q9KL81 | |
| | | | | Q9KPA0 | |
| | | | | Q9KL73 | |
| | | | | Q9KNG0 | |
| | | | | Q9KSJ4 | |
| | | | | Q9KPZ1 | |
| | | | | Q9KNI6 | |
| | | | | Q9KVT0 | |
| | | | | Q9KST0 | |

Based on antigenicity, allergenicity, toxicity B-cell and T-cell epitope prediction, there are 20 proteins (potential vaccine candidates) that are predicted to elicit immunogenic response in human. In addition, we have also predicted 20 potential drug targets that are unanimously predicted to be virulent by 5 approaches (amino acid composition, Dipeptide composition, PSI-BLAST created PSSM profiles, cascade of SVMs and PSI-BLAST and higher order dipeptide composition) of VirulentPred.

allergenic properties, and low toxicity, indicating they are potential candidates for vaccine development. Each of the uncharacterized proteins possesses linear as well as discontinuous epitopes. The BcePred, a server for predicting linear B-cell epitopes, was used to predict epitopes for 24 proteins with unknown characteristics. Proteins containing more than 100 linear B-cell epitopes include Q9KRD2 (491), Q9KVG3 (432), Q9KT38 (294), Q9KLK5 (148), Q9KKL8 (222), Q9KSV6 (123), Q9KND3 (106), Q9KND9 (100), all of which have an epitope

length of three amino acid residues and a score greater than 0.5 (**S12 Table** and **S6 Fig**). Discontinuous B-cell epitopes were predicted using DiscoTope tools, with a threshold discotope propensity score of -3.7. The top proteins with the high number of discontinuous epitopes were Q9KRD2 (45), Q9KVG3 (186), Q9KT38 (57), Q9KKL8 (125), Q9KLK5 (70), Q9KND9 (59), Q9KU58 (75), B1B1N2 (55) and Q9KL81 (48) (**S13 Table** and **S7 Fig**).

## T-cell epitopes identification

The NetCTL servers were used to predict the T-cell epitopes of the proteins. CTL epitopes for the candidate protein are predicted using NetCTL 1.2 based on MHC class-I binding capacity, TAP transport efficiency, and proteasomal C-terminal cleavage [39]. All three predictions' scores were added together, and the resulting merged score of all three predictions was used to determine the threshold for CTL epitope identification, which was set at 0.75 (**S14 Table**). Helper T-lymphocyte (HTL) epitopes were identified using the IEDB MHCII server using Human/HLA-DR as the species/locus and 7 alleles of human leukocyte antigen (HLA) (**S15 Table**). High MHC II affinity was found when the percentile rank was matched to the SwissProt database. Based on MHC-I and MHC-II binding of candidate proteins, 20 proteins were screened out of the pool of uncharacterized proteins to have T-cell epitopes (**Table 2**).

## Conclusion

Understanding the role of uncharacterized and hypothetical proteins in bacteria are essential for bridging the gaps in our knowledge of gene functions, interactions and molecular mechanisms leading to bacterial pathogenesis. In this study, we analyzed 51 uncharacterized and hypothetical proteins of *V. cholerae* whose expression is altered in CgtA-depleted conditions, as shown by transcriptomic and proteomic studies. We determined the physicochemical properties of the strains, such as molecular weight, theoretical isoelectric point, extinction coefficient, instability index, aliphatic index, grand average of hydropathicity (GRAVY), and total number of negatively and positively charged residues. The molecular weight of the proteins ranged from 92 kDa to 5 kDa, and the theoretical pI ranged from 4.19 to 11.40, with the majority of the proteins being acidic. The theoretical pI is defined as the point at which a particular molecule carries no net electrical charge at the pH scale and is useful for understanding protein charge stability. We also computed the solubilities, subcellular localizations and probable functions of the proteins and identified their domains and families using various bioinformatics tools and databases. We have observed that several of these uncharacterized proteins are involved in essential cellular processes associated with cholera pathogenesis like transcription, translation, phosphorelay signal transduction, motility and chemotaxis. These predictions are very crucial and effective for hypothesis generation and designing experiments for further validation. For functional protein association networks, STRING was used for the prediction of interactions between our uncharacterized candidate proteins and other partners. Characterizing protein–protein interactions is vital to reinforce our understanding of protein function and the biology of the cell. Additionally, we employed reverse vaccinology and immunoinformatic approach to identify potential vaccine candidates and potential drug targets that will pave the way towards novel drug discovery and vaccine design against cholera pathogen. We also constructed 2D and 3D structural models with PSIPRED and Alphafold2-mmseqs2 respectively, which were further validated with ProSA and PROCHECK. The *in-silico* studies were further validated by experimental studies by cloning and expression of two crucial proteins (potential vaccine candidates) whose expression was altered in CgtA-depleted conditions. The protein Q9KRD2 is a 92 kDa EAL and sensory PAS domain-containing putative phosphodiesterase that is downregulated when *cgtA* is knocked down in the *V. cholerae* genome. In

contrast, proteomic studies have shown that the 12 kDa protein Q9KU58 is upregulated when *cgtA* is knocked down in the *V. cholerae* genome. Our findings based on in-depth quantitative computational analysis and experimental work will help us to understand the biology of cholera pathogenesis as a whole, and also identify potential therapeutic leads at the molecular level.

## Supporting information

**S1 Table. RNA-seq data depicting the expression of genes that were downregulated in *cgtA* knockdown strain of *Vibrio cholerae*: The p-value and fold change of each protein are recorded and tabulated indicating the alteration in the expression of each protein when *cgtA* is knocked down from *V. cholerae* genome.**
(DOCX)

**S2 Table. Physicochemical properties of the 51 uncharacterized proteins: Physicochemical properties like molecular weight, isoelectric point, aliphatic index(thermostability), GRAVY (solubility), molar extinction coefficient, instability index (Stability in solution) and number of positive and negative residues were assessed.**
(DOCX)

**S3 Table. Amino acid composition (%) of the uncharacterized and hypothetical proteins.**
(DOCX)

**S4 Table. Subcellular localization of the uncharacterized proteins: - The subcellular location of 51 uncharacterized proteins were assessed using PSORTb and PSLPred.**
(DOCX)

**S5 Table. Identification of domains of 51 uncharacterized proteins using InterPro, SMART and PROSITE.**
(DOCX)

**S6 Table. Prediction of soluble and transmembrane protein and determination of transmembrane region present within the uncharacterized proteins.**
(DOCX)

**S7 Table. Prediction of Protein Function using PFP (Protein Function Prediction) Server: The molecular function, biological function and cellular location was predicted for each of the proteins using Protein Function Prediction (PFP) server as depicted by PFP score.**
(DOCX)

**S8 Table. Prediction of Protein Function using Argot2 Server: The molecular function, biological function and cellular location was predicted for each of the proteins using Argot2 server as depicted by Argot2 score.**
(DOCX)

**S9 Table. Protein-protein interaction: Identification of interacting protein partners of candidate uncharacterized protein by STRING.**
(DOCX)

**S10 Table. Antigenicity, allergenicity and toxicity of candidate uncharacterized proteins.**
(DOCX)

**S11 Table. Prediction of virulence of the candidate protein by VirulentPred.**
(DOCX)

**S12 Table. Identification of linear B-cell epitopes present within the candidate proteins.**
(DOCX)

**S13 Table. Identification of discontinuous B-cell epitopes present within the candidate uncharacterized proteins.**
(DOCX)

**S14 Table. Prediction of MHC-I binding of candidate proteins by NetCTL.**
(DOCX)

**S15 Table. Prediction of binding affinity between MHC-II and candidate uncharacterized protein depicted by scores generated from IEDB MHC II.**
(DOCX)

**S1 Fig. Hydropathy plot (Kyte/Doolittle plot): Hydropathy plots illustrating the hydrophilic and hydrophobic region within each of the uncharacterized proteins.**
(DOCX)

**S2 Fig. Computational prediction of protein-protein interaction (PPI) network by STRING.**
(DOCX)

**S3 Fig. Identification of secondary structures present within each of the uncharacterized protein by PSIPRED.**
(DOCX)

**S4 Fig. Prediction of disordered regions present within candidate uncharacterized proteins by DISOPRED.**
(DOCX)

**S5 Fig. Construction of 3-dimensional structures of uncharacterized proteins by ALpha-FOLD-mmseq2 and validation of the structures.**
(DOCX)

**S6 Fig. Graphical representation of linear B-cell epitope prediction for 24 uncharacterized proteins.**
(DOCX)

**S7 Fig. Graphical representation of discontinuous B-cell epitope present within 24 uncharacterized proteins.**
(DOCX)

## Author Contributions

**Conceptualization:** Partha Pratim Datta.

**Data curation:** Sritapa Basu Mallick.

**Formal analysis:** Sritapa Basu Mallick, Sagarika Das, Aravind Venkatasubramanian, Sourabh Kundu.

**Investigation:** Sritapa Basu Mallick, Sagarika Das.

**Methodology:** Sritapa Basu Mallick, Partha Pratim Datta.

**Software:** Sritapa Basu Mallick.

**Supervision:** Partha Pratim Datta.

**Validation:** Sritapa Basu Mallick, Aravind Venkatasubramanian.

**Writing – original draft:** Sritapa Basu Mallick.

**Writing – review & editing:** Sritapa Basu Mallick, Partha Pratim Datta.

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
