## [Decision Letter · Decision Letter 0]

16 Aug 2024

PONE-D-24-29986Comprehensive in silico analyses of fifty-one uncharacterized proteins from Vibrio choleraePLOS ONE

Dear Dr. DATTA,

Thank you for submitting your manuscript to PLOS ONE. After careful consideration, we feel that it has merit but does not fully meet PLOS ONE’s publication criteria as it currently stands. Therefore, we invite you to submit a revised version of the manuscript that addresses the points raised during the review process.

We look forward to receiving your revised manuscript.

Kind regards,

Rajesh Kumar Pathak, Ph.D.

Academic Editor

PLOS ONE

Journal Requirements:

3. Please include your tables as part of your main manuscript and remove the individual files. Please note that supplementary tables (should remain/ be uploaded) as separate "supporting information" files

**Additional Editor Comments:**

The manuscript has been reviewed and is found to be interesting. However, the reviewers have raised some queries that need to be addressed in the revised manuscript.

Reviewers' comments:

Reviewer's Responses to Questions

**Comments to the Author**

1. Is the manuscript technically sound, and do the data support the conclusions?

Reviewer #1: Yes

Reviewer #2: Yes

Reviewer #3: Partly

2. Has the statistical analysis been performed appropriately and rigorously? 

Reviewer #1: Yes

Reviewer #2: Yes

Reviewer #3: Yes

3. Have the authors made all data underlying the findings in their manuscript fully available?

Reviewer #1: Yes

Reviewer #2: Yes

Reviewer #3: Yes

4. Is the manuscript presented in an intelligible fashion and written in standard English?

Reviewer #1: Yes

Reviewer #2: Yes

Reviewer #3: No

5. Review Comments to the Author

**Reviewer #1: **The authors have tried to annotate a list of uncharacterized proteins from Vibrio cholerae and analyzed their potency for vaccine development. I find the work is very much useful for the scientific community.

**Reviewer #2:** In the manuscript “Comprehensive in silico analyses of fifty-one uncharacterized proteins from Vibrio cholerae", submitted by Mallick et al., the authors present an intriguing study on the characterization of 51 uncharacterized and hypothetical proteins in Vibrio cholerae. The focus on understanding these proteins' physicochemical properties, functional associations, and potential roles is well-justified and contributes to the broader understanding of cholera biology. However, a few areas could be further developed to enhance the clarity and impact of the paper.

• The introduction could benefit from a more detailed explanation of the significance of these particular proteins. Providing a stronger rationale for their selection and discussing how this study advances our understanding of V. cholerae would help set the stage for your findings.

• Your methodological approach is comprehensive, especially in describing the physicochemical properties of the proteins. However, it would be useful to elaborate on why these specific properties are important for studying hypothetical proteins. Additionally, more details on the tools and databases used for solubility predictions, subcellular localization, and domain identification would enhance reproducibility and transparency.

• The use of STRING for predicting protein-protein interactions is a strong aspect of the study, but the manuscript could be improved by providing more insight into these predicted interactions. Discussing how these interactions contribute to understanding the proteins' potential functions would add depth to your analysis.

• The experimental validation of two proteins is a valuable part of the study, but it would be beneficial to expand on the results of these experiments. Discuss how the experimental findings align with or differ from your in-silico predictions, as this correlation could be a significant strength of the paper.

• In terms of structure, the manuscript could be improved with smoother transitions between sections, particularly when moving from computational analysis to experimental validation. This would help create a more cohesive narrative. Additionally, the conclusion could be expanded to emphasize the broader significance of your findings and suggest possible directions for future research.

• The manuscript would benefit from grammatical revisions for clarity, and some technical terms should be defined more clearly for readers who may not be specialists in the field. Including figures or tables to summarize key findings, such as the physicochemical properties or interaction networks, would also enhance the readability and impact of the paper.

• Overall, the study presents valuable findings that contribute to the understanding of V. cholerae proteins but addressing these suggestions could strengthen the manuscript and make it more accessible and impactful.

**Reviewer #3: **1. Explanation needed to choose CgtA over another available drug targets.

2. Explain significance of predicted 20 drug targets.

3. Explain relationship between 20 predicted candidate vaccines and 20 predicted drug targets.

4. Why only 2 uncharacterized proteins were selected for validation of 20 predicted candidate vaccines. Study should include more proteins for validation.

5. Abstract should be re-written, and manuscript should be checked to improve English.

6. PLOS authors have the option to publish the peer review history of their article (what does this mean?). If published, this will include your full peer review and any attached files.

Reviewer #1: **Yes: **Rajabrata Bhuyan

Reviewer #2: No

Reviewer #3: No

---

## [Author Response · Author response to Decision Letter 0]

3 Sep 2024

Response to reviewers

1. Reviewer #1: The authors have tried to annotate a list of uncharacterized proteins from Vibrio cholerae and analysed their potency for vaccine development. I find the work is very much useful for the scientific community.

Author’s response: Thank you so much for your kind comment.

2. Reviewer #2: In the manuscript “Comprehensive in-silico analyses of fifty-one uncharacterized proteins from Vibrio cholerae", submitted by Mallick et al., the authors present an intriguing study on the characterization of 51 uncharacterized and hypothetical proteins in Vibrio cholerae. The focus on understanding these proteins' physicochemical properties, functional associations, and potential roles is well-justified and contributes to the broader understanding of cholera biology. However, a few areas could be further developed to enhance the clarity and impact of the paper.

 (A) The introduction could benefit from a more detailed explanation of the significance of these particular proteins. Providing a stronger rationale for their selection and discussing how this study advances our understanding of V. cholerae would help set the stage for your findings.

Author’s response: The lines 58-67 (Page 2-3) have been modified. The following lines (highlighted in yellow) are added within the manuscript: -

“Understanding these biochemically, structurally and functionally uncharacterized proteins can pave the way towards mechanistic insights of how this essential GTPase, CgtA, exerts its pleotropic effect. Hence, we carried out a comprehensive in silico analysis of these uncharacterized proteins, which allowed us to predict their physicochemical and immunogenic properties and hypothesize and design various in vitro and in vivo experiments to characterize these proteins, which will lead us to understand the basis of cholera pathogenesis at a deeper level. Also, we have successfully identified a number of potential drug-targets and vaccine candidates among those 51 uncharacterized proteins that will facilitate the production of various vaccine constructs and drugs through in vivo and in vitro experiments against the Vibrio cholerae pathogen.”

(B) Your methodological approach is comprehensive, especially in describing the physicochemical properties of the proteins. However, it would be useful to elaborate on why these specific properties are important for studying hypothetical proteins. Additionally, more details on the tools and databases used for solubility predictions, subcellular localization, and domain identification would enhance reproducibility and transparency.

Author’s response: Lines 115-136 (Page 4-5) have been modified to include more information as per the suggestion. 

“The ProtParam web server by ExPASy (https://web.expasy.org/protparam/) was used to identify the physicochemical characteristics of the uncharacterized proteins such as molecular weight, theoretical isoelectric point (pI), amino acid composition profile (%), molar extinction coefficient, instability index, aliphatic index, grand average hydropathy (GRAVY), and the total number of positively charged (Arg+Lys) and negatively charged (Asp+Glu) residues based on their amino acid sequences (Supplementary Table 2, Supplementary Table 3, Fig. 2A). The molar extinction coefficient is the measure of the amount of light that proteins absorb at a specific wavelength. A high molar extinction coefficient value indicates the presence of a high concentration of cysteine, tryptophan, and tyrosine in the candidate proteins. The instability index provides an estimation of the stability of a protein in a test tube. A protein whose instability index is greater than 40 is predicted to be instable in solution. The aliphatic index of a protein is defined as the relative volume occupied by aliphatic side chain amino acids like alanine, valine, leucine, and isoleucine. It may be considered as a positive factor for the enhancement of thermo-stability of globular protein. A high aliphatic index indicates that a protein is thermo-stable over a wide temperature range. The GRAVY score for a protein is calculated as the sum of the hydropathy values of all of the amino acids divided by the number of residues in the query sequence. A low GRAVY value indicates the possibility of a protein being a globular or hydrophilic protein rather than membranous. A comprehensive analysis of these physicochemical properties will provide us an insight of the possible biological functions of these uncharacterized proteins. In addition, the predicted traits and properties like instability index and solubility will allow us to design effective strategies to express and purify these proteins for downstream biochemical and functional characterization.”

Lines 143-145 (Page 5) were added to describe PSORTb tool which is used to predict subcellular localization of a protein. 

“A score is assigned for every localization site which reflects the confidence level of the prediction. A score, higher than the cut-off value 7.5 indicates a strong confidence in the predicted localization.”

We have also added a minor detail about the tool PSLPred in line 147-149 (Page 5). The following line was modified as follows: -

“Further, the results were confirmed by PSLpred (https://webs.iiitd.edu.in/raghava/pslpred/submit.html), which accurately predicts the subcellular localization of uncharacterized proteins based on a hybrid approach that integrates PSI_BLAST and three SVM based on physicochemical properties, residue composition and dipeptide.”

Lines 157-162 (Page 5-6) was modified and added in the manuscript file. The following lines were modified and added where we have described the tools used for domain identification.

“The results generated by the “Interpro” database were validated using SMART (http://smart.embl-heidelberg.de/), a tool that extensively annotates a protein domain based on functional class, phyletic distributions, tertiary structures and functionally important residues. We further validated the results using PROSITE (https://prosite.expasy.org/), a large database of protein domains, families, and functional sites that identifies conserved sequences and functional site within candidate proteins which is crucial for understanding their structure and function” 

(C) The use of STRING for predicting protein-protein interactions is a strong aspect of the study, but the manuscript could be improved by providing more insight into these predicted interactions. Discussing how these interactions contribute to understanding the proteins' potential functions would add depth to your analysis.

Author’s response: Lines 386-397 (Page 13) were added to explain the protein-protein interaction network (PPI) derived from STRING version 11.5. The following lines were added in the manuscript: -

“Identification of protein-protein interaction network

Protein-protein interaction or PPI network for 51 uncharacterized proteins were visualized in STRING version 11.5, which predicts the interactive protein partners using a confidence score above 0.7. The interacting partners of each of the uncharacterized proteins are tabulated in Supplementary material 3, Table S15. The highest confidence score of 0.996 and 0.994 were observed for the interaction between SpoVR family protein, Q9KQX3 and a hypothetical protein, Q9KQX4 (VC_1873) and uncharacterized protein, Q9KQX5 (VC_1872) respectively. Other interaction with high confidence score of 0.993 was seen in PPI between EAL domain containing Q9KRD2 and Q9KPJ7 (Gene name- VC_2370) which is a sensory box containing diguanylate cyclase enzyme. The findings in this study will allow us to reinforce our understanding on the probable functions of these uncharacterized proteins and decipher the molecular pathways involved with cellular functions.”

(D) The experimental validation of two proteins is a valuable part of the study, but it would be beneficial to expand on the results of these experiments. Discuss how the experimental findings align with or differ from your in-silico predictions, as this correlation could be a significant strength of the paper.

Author’s response: Lines 428-443 (Page 14) were added where we have compared the predicted solubility and molecular weight of the selected proteins with the experimental results. The following lines were added (as highlighted in the manuscript):-

 “The GRAVY index for Q9KRD2 and Q9KU58 were predicted to be -0.172 and -0.319, respectively, which indicates their solubility in solution. Also, the predictions based on SOSUI and DeepTMHMM, the two proteins were predicted to be soluble. However, while expressing the proteins in the heterologous system of E. coli BL21 (pLySs) both the proteins showed solubility issues. We have used mild concentration (2M) of polar reagent, urea, to solubilize the inclusion body. The solubilized inclusion body were then purified using a nickel-NTA column. Finally, the expression of the purified protein Q9KRD2 (Figure 6G) and Q9KU58 (Figure 6I) were checked by SDS‒PAGE, respectively. The purified Q9KRD2 showed a clear band near 100 kDa protein marker and purified Q9KU58 showed a distinct band near 15 kDa band as expected from in silico analysis. Further, the expression of the two purified protein were validared western blot analysis (Figure 6H and J).

The 3-dimensional model structure of Q9KRD2 and Q9KU58 is shown in (Figure 6K and 6P). Ramachandran plot validation for the protein Q9KRD2 suggested that 91.70% the residues were seen to be in the most favoured region (Figure 6L and 6M) while for Q9KU58 79.30% (Figure 6Q and 6R). Additionally, ProSA z-score of Q9KRD2 and Q9KU58 found as -13.31 (Figure 6N) and -2.66 (Figure 6S) respectively ProSA was also used for assessing local model quality of Q9KRD2 (Figure 6O) and Q9KU58 (Figure 6T).” 

Other aspects such as domain characterization, subcellular localization, protein-protein interactions and structure are beyond the scope of this manuscript. However, we are currently exploring these aspects experimentally which we would like to present it in a separate manuscript.

(E) In terms of structure, the manuscript could be improved with smoother transitions between sections, particularly when moving from computational analysis to experimental validation. This would help create a more cohesive narrative. Additionally, the conclusion could be expanded to emphasize the broader significance of your findings and suggest possible directions for future research.

Author’s response: The “Result and discussion” part describing the structure of the 51 uncharacterized proteins, and the experimental validation of two proteins, Q9KRD2 and Q9KU58 by cloning and expression has been updated as shown by the highlighted region in page-13 and 14. We have also modified the conclusion as shown by the highlighted region in page 17-18 of manuscript file. Following is the conclusion that we have added in the manuscript.

“Understanding the role of uncharacterized and hypothetical proteins in bacteria are essential for bridging the gaps in our knowledge of gene functions, interactions and molecular mechanisms leading to bacterial pathogenesis. In this study, we analyzed 51 uncharacterized and hypothetical proteins of V. cholerae whose expression is altered in CgtA-depleted conditions, as shown by transcriptomic and proteomic studies. We determined the physicochemical properties of the strains, such as molecular weight, theoretical isoelectric point, extinction coefficient, instability index, aliphatic index, grand average of hydropathicity (GRAVY), and total number of negatively and positively charged residues. The molecular weight of the proteins ranged from 92 kDa to 5 kDa, and the theoretical pI ranged from 4.19 to 11.40, with the majority of the proteins being acidic. The theoretical pI is defined as the point at which a particular molecule carries no net electrical charge at the pH scale and is useful for understanding protein charge stability. We also computed the solubilities, subcellular localizations and probable functions of the proteins and identified their domains and families using various bioinformatics tools and databases. We have observed that several of these uncharacterized proteins are involved in essential cellular processes associated with cholera pathogenesis like transcription, translation, phosphorelay signal transduction, motility and chemotaxis. These predictions are very crucial and effective for hypothesis generation and designing experiments for further validation. For functional protein association networks, STRING was used for the prediction of interactions between our uncharacterized candidate proteins and other partners. Characterizing protein–protein interactions is vital to reinforce our understanding of protein function and the biology of the cell. Additionally, we employed reverse vaccinology and immunoinformatic approach to identify potential vaccine candidates and potential drug targets that will pave the way towards novel drug discovery and vaccine design against cholera pathogen. We also constructed 2D and 3D structural models with PSIPRED and Alphafold2-mmseqs2 respectively, which were further validated with ProSA and PROCHECK. The in-silico studies were further validated by experimental studies by cloning and expression of two crucial proteins (potential vaccine candidates) whose expression was altered in CgtA-depleted conditions. The protein Q9KRD2 is a 92 kDa EAL and sensory PAS domain-containing putative phosphodiesterase that is downregulated when cgtA is knocked down in the V. cholerae genome. In contrast, proteomic studies have shown that the 12 kDa protein Q9KU58 is upregulated when cgtA is knocked down in the V. cholerae genome. Our findings based on in-depth quantitative computational analysis and experimental work will help us to understand the biology of cholera pathogenesis as a whole, and also identify potential therapeutic leads at the molecular level.”

(F) The manuscript would benefit from grammatical revisions for clarity, and some technical terms should be defined more clearly for readers who may not be specialists in the field. Including figures or tables to summarize key findings, such as the physicochemical properties or interaction networks, would also enhance the readability and impact of the paper.

Author’s response: The entire manuscript was checked in English language editing tool, viz., Grammarly and Curie, for the purpose of enhancement of clarity for readers. Furthermore, the result and discussion part, especially the reverse vaccinology and immunoinformatic part have been explained in greater details for better clarity. The figure and table captions were also re-written elaborately.

(G) Overall, the study presents valuable findings that contribute to the understanding of V. cholerae proteins but addressing these suggestions could strengthen the manuscript and make it more accessible and impactful.

Author’s response: Thank you so much for your valuable input and suggestions. We tried our level best to respond to your insightful queries. Thank you.

3. Reviewer #3: 

(A)Explanation needed to choose CgtA over another available drug targets.

Author’s response: 

Although CgtA is a multifunctional essential GTPase in Vibrio cholerare, not much is known about its mechanisms of actions. Hence our laboratory has been working on it to better understand its functionality and find new or alternate drug targets based on CgtA research in V. cholerare. 

(B)Explain significance of predicted 20 drug targets.

Author’s response: Virulent proteins play a monumental role in pathogenesis of infectious disease and can be targeted for drug-design and therapeutic interventions. 20 potential drug targets were predicted from a pool of 51 uncharacterized proteins based on assessment generated by VirulentPred which exploits 5 prediction approaches: - 

(a) Amino acid composition, 

(b) Dipeptide composition, 

(c) PSI-BLAST generated PSSM profiles, 

(d) A cascade of SVMs and PSI-BLAST, and 

(e) Higher order dipeptide composition.

The threshold value for each approach was taken as 0 (zero). A positive score indicates that the protein is virulent. Whereas, a negative value indicates that the protein is a non-virulent protein. Understanding the functi

---

## [Decision Letter · Decision Letter 1]

17 Sep 2024

Comprehensive in silico analyses of fifty-one uncharacterized proteins from Vibrio cholerae

PONE-D-24-29986R1

Dear Dr. DATTA,

We’re pleased to inform you that your manuscript has been judged scientifically suitable for publication and will be formally accepted for publication once it meets all outstanding technical requirements.

Kind regards,

Rajesh Kumar Pathak, Ph.D.

Academic Editor

PLOS ONE

Additional Editor Comments (optional):

The authors have addressed the comments and revised the manuscript, making it acceptable now.

Reviewers' comments:

Reviewer's Responses to Questions

**Comments to the Author**

1. If the authors have adequately addressed your comments raised in a previous round of review and you feel that this manuscript is now acceptable for publication, you may indicate that here to bypass the “Comments to the Author” section, enter your conflict of interest statement in the “Confidential to Editor” section, and submit your "Accept" recommendation.

Reviewer #3: All comments have been addressed

Reviewer #4: All comments have been addressed

2. Is the manuscript technically sound, and do the data support the conclusions?

Reviewer #3: Partly

Reviewer #4: (No Response)

3. Has the statistical analysis been performed appropriately and rigorously? 

Reviewer #3: Yes

Reviewer #4: N/A

4. Have the authors made all data underlying the findings in their manuscript fully available?

Reviewer #3: Yes

Reviewer #4: Yes

5. Is the manuscript presented in an intelligible fashion and written in standard English?

Reviewer #3: Yes

Reviewer #4: Yes

6. Review Comments to the Author

Reviewer #3: Authors have made significant correction to improve their manuscript and i am partially satisfied with their explanation.

Reviewer #4: I am grateful to the authors for carefully considering all the comments. I hope this article will have a positive effect on the world of science

7. PLOS authors have the option to publish the peer review history of their article (what does this mean?). If published, this will include your full peer review and any attached files.

Reviewer #3: No

Reviewer #4: No

---

## [Editor Report · Acceptance letter]

25 Sep 2024

PONE-D-24-29986R1 

PLOS ONE

Dear Dr. DATTA, 

I'm pleased to inform you that your manuscript has been deemed suitable for publication in PLOS ONE. Congratulations! Your manuscript is now being handed over to our production team.

Kind regards, 

on behalf of

Dr. Rajesh Kumar Pathak 

Academic Editor

PLOS ONE